# ECHO FLOW NETWORKS
# WITH INFINITE-HORIZON MEMORY

## ABSTRACT

At the heart of time-series forecasting (TSF) lies a fundamental challenge: how can models efficiently and effectively capture long-range temporal dependencies across ever-growing sequences? While deep learning has brought notable progress, conventional architectures often face a trade-off between computational complexity and their ability to retain accumulative information over extended horizons.

Echo State Networks (ESNs), a class of reservoir computing models, have recently regained attention for their exceptional efficiency, offering constant memory usage and per-step training complexity regardless of input length. This makes them particularly attractive for modeling extremely long-term event history in TSF. However, traditional ESNs fall short of state-of-the-art performance due to their limited nonlinear capacity, which constrains both their expressiveness and stability.

We introduce ECHO FLOW NETWORKS (**EFNs**), a framework composed of a group of extended Echo State Networks (**X-ESNs**) with MLP readouts, enhanced by our novel Matrix-Gated Composite Random Activation (**MCRA**), which enables complex, neuron-specific temporal dynamics, significantly expanding the network's representational capacity without compromising computational efficiency. In addition, we propose a dual-stream architecture in which recent input history dynamically selects signature reservoir features from an infinite-horizon memory, leading to improved prediction accuracy and long-term stability.

Extensive evaluations on five benchmarks demonstrate that **EFNs** achieves up to 4× faster training and 3× smaller model size compared to leading methods like PatchTST, reducing forecasting error from 43% to 35%, a 20% relative improvement. One instantiation of our framework, **EchoFormer**, consistently achieves new state-of-the-art performance across five benchmark datasets: ETTh, ETTm, DMV, Weather, and Air Quality.

## 1 INTRODUCTION

Time-series forecasting (TSF) is a fundamental problem at the core of scientific discovery and decision-making, powering critical applications in climate science, finance, healthcare, and energy systems by predicting future trends from historical data. A key challenge in TSF is modeling *long-range temporal dependencies*, where the effects of past events unfold gradually. For instance, seasonal droughts can influence wildfire risk months later, and early market signals may foreshadow shifts in cryptocurrency prices (Chen et al., 2023; Ma et al., 2024).

While capturing long-range dependencies, and ideally learning from the entire history of inputs, is essential for accurate prediction, fully leveraging such historical context remains an open challenge due to two critical limitations. First, **computational inefficiency**: Transformer-based models like PatchTST (Nie et al., 2023) exhibit quadratic complexity in time and memory (Vaswani et al., 2017), making them impractical for ultra-long sequences. Even with recent efficiency improvements (Jia et al., 2024; Lin et al., 2023; Chi, 2024), these models still rely on backpropagation through time, resulting in high training cost and limited scalability. Second, **model limitations**: many models struggle with vanishing gradients and suffer from bifurcation issues, leaving significant room for performance improvement using long history (Lim & Zohren, 2021; Zhou et al., 2021; Nie, 2023). (Lim & Zohren, 2021; Zhou et al., 2021; Nie, 2023).

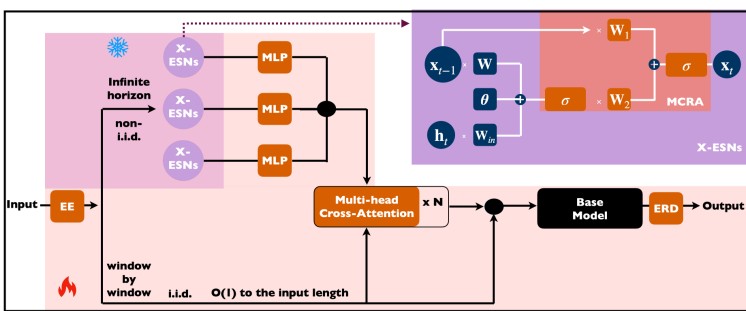

Figure 1: **EFNs** Framework with X-ESNs, and **MCRA** ($\mathbf{W}_1$, $\mathbf{W}_2$, $\sigma$).

**Echo State Networks (ESNs)** (Jaeger, 2001; Lukoševičius & Jaeger, 2009), as a kind of reservoir computing, offer a promising yet underexplored alternative. ESNs are lightweight autoregressive models with **linear time** and **constant space complexity**. Unlike attention-based models that truncate historical inputs or RNNs that compress past information into fixed-size hidden states, ESNs update their internal dynamics in a streaming fashion using randomly initialized, fixed weights, **without backpropagation**. This design enables a constant $\mathcal{O}(1)$ training cost per time step and a constant $\mathcal{O}(1)$ memory footprint, as well as an overall training time complexity of $\mathcal{O}(N)$. Furthermore, the reservoir acts as a denoising temporal encoder that is robust to bifurcations and gradient decay (Bollt, 2021; Vlachas et al., 2020), making ESNs ideal for modeling long-sequence patterns.

Yet, despite their theoretical advantages, ESNs have historically underperformed on time-series forecasting (TSF) tasks and, as a result, are not widely adopted. In this work, we identify three core limitations contributing to this underperformance and propose solutions to overcome them. First, ESNs suffer from **limited expressiveness in both state updates and readouts**. Classical ESNs typically employ a single nonlinearity (e.g., `tanh`) in the state update and a linear readout layer, restricting their ability to capture complex temporal dynamics. This limitation is especially pronounced when the reservoir is too small or poorly aligned with task-relevant features, leading to failures in modeling hierarchical, compositional, or multi-scale structures that are challenging to decode linearly. Second, ESNs are **highly sensitive to random initialization**: since reservoir weights are fixed and randomly assigned rather than learned, performance can vary significantly between runs, often necessitating heuristic tuning or task-specific adjustments for stability (Rodan & Tiňo, 2011; Lu et al., 2017). Third, ESNs **lack a dynamic, token-specific weighting mechanism**, such as attention, which hinders their ability to selectively focus on informative inputs, adapt to abrupt temporal shifts, or model non-smooth and discrete patterns. Together, these limitations constrain the ability of ESNs to achieve state-of-the-art performance.

In this paper, we propose **EFNs** (**ECHO FLOW NETWORKS**), a novel framework that combines the efficiency of classical Echo State Networks (ESNs) with the expressive power of modern deep sequential models. **EFNs** enhances ESNs with spiking dynamics, grouped Extended ESNs (X-ESNs), and a dual-stream fusion mechanism for expressiveness, stability, and near token dependency.

As illustrated in Figure 1, **EFNs** consists of two main components integrated via masked multi-head attention: a recent input encoder operating over the $k$-length short-term window, and a **grouped X-ESN** that captures dynamics from the entire input history. **EFNs** can function as a standalone forecasting model or be used to boost existing TSF methods in a black-box manner. Specifically, our approach introduces four key innovations:

1. **X-ESNs with MCRA for Expressiveness:** Extended ESNs (**X-ESNs**) augment traditional Echo State Networks (ESNs) with our proposed Matrix-Gated Composite Random Activation (**MCRA**). Unlike the standard single `tanh` activation, **MCRA** employs a composition of two nonlinear activation functions to jointly transform the current input and the previous state, enabling the modeling of more complex temporal structures. Each activation function is randomly selected from a predefined set (e.g., `ReLU`, `Leaky ReLU`, `tanh`, and `Sigmoid`), increasing functional diversity within the ensemble mechanism of **Group X-ESNs** for improved accuracy and stability.

Furthermore, **MCRA** replaces the scalar leaky integration parameter with matrix-valued gates, allowing for more expressive and adaptable neuron-specific temporal dynamics.

The conventional ESNs, i.e., LI-ESN (Jaeger et al., 2007), evolves the reservoir state $\mathbf{x}_t \in \mathbb{R}^{N_r}$ as:

$$\mathbf{x}_t = (1 - \alpha)\mathbf{x}_{t-1} + \alpha \tanh(\mathbf{W}_{\text{in}}\mathbf{h}_t + \boldsymbol{\theta} + \mathbf{W}\mathbf{x}_{t-1}), \tag{1}$$

where $\mathbf{h}_t \in \mathbb{R}^{N_u}$ is the input at time $t$, $\mathbf{W}_{\text{in}}, \mathbf{W}$, and $\boldsymbol{\theta}$ are the input, recurrent, and bias parameters, respectively, and $\alpha \in [0, 1]$ is the leaky integration rate controlling memory decay. $N_r$ and $N_u$ are reservoir and input dimension, respectively. Weights are drawn from $\mathcal{U}[-\sigma_{\text{in}}, \sigma_{\text{in}}]$, and $\mathbf{W}$ is scaled to satisfy the Echo State Property (ESP) (Tiňo et al., 2007) via the spectral radius condition.

We extend this formulation in **X-ESNs** by introducing the *Matrix-Gated Composite Random Activation* (**MCRA**), resulting in the following update:

$$\mathbf{x}_t = \sigma_2 \left( \mathbf{W}_1 \mathbf{x}_{t-1} + \mathbf{W}_2 \, \sigma_1 \left( \mathbf{W}_{\text{in}} \mathbf{h}_t + \boldsymbol{\theta} + \mathbf{W}_0 \mathbf{x}_{t-1} \right) \right),$$

where $\mathbf{W}_1$ and $\mathbf{W}_2$ are matrix-valued gates constrained by normalization, and $\sigma_1, \sigma_2$ are nonlinear activation functions randomly selected from a predefined set (e.g., `tanh`, `ReLU`, `sigmoid`).

The **MCRA** mechanism introduces four key novel elements:

- **Nonlinear Activation**: emphasizes that nonlinear transformations are applied both before and after the reservoir update, increasing expressiveness;
- **Matrix-Gated**: leaky integration is generalized using matrix-valued gates (replacing scalar $\alpha$), allowing for more flexible and expressive dynamics;
- **Composite**: the activation is formed as a nested composition of two nonlinearities, enabling deeper feature transformations;
- **Randomized**: the nonlinear functions $\sigma_1$ and $\sigma_2$ are chosen randomly, encouraging diversity across the network ensemble.

This formulation allows for neuron-specific, nontrivial temporal dynamics and significantly enhances the representational capacity of the reservoir. The classical ESNs are recovered as a special case when $\sigma_2$ is the identity function and $\mathbf{W}_1, \mathbf{W}_2$ reduce to scalar weights.

2. **Heterogeneous Group X-ESNs for Stability:** To address the sensitivity of ESNs to random initialization, we propose **Group X-ESNs**: ensembles of independently initialized X-ESN units whose outputs are aggregated to produce a stable, low-variance memory stream. Specifically, we employ a heterogeneous group of X-ESNs, each with randomly assigned pair of activation functions and varying dimensions, to promote diversity. This approach reduces performance variance and improves robustness without compromising efficiency.

3. **Recurrent Dual-Stream for Token Selection:** We design a dual-stream architecture that combines Group **X-ESNs** (for long-range, non-i.i.d. dependencies) with a short-context base TSF model (e.g., PatchTST) trained on local, i.i.d. patterns. A cross-attention readout enables token-wise alignment, allowing the model to selectively attend to relevant historical states. This fusion effectively captures both persistent trends and local variations. Since the base model operates on a fixed-length window and **X-ESNs** are untrained reservoirs (no backpropagation), the overall training complexity remains linear in sequence length and constant per time step, with only the MLP readout layer and the cross-attention combiner trained via backpropagation.

4. **Standalone or Model Booster: EFNs** is a modular framework designed to perform time-series forecasting (TSF) either as a standalone model or as an enhancement for any baseline model, regardless of its architecture. We instantiate and evaluate multiple variants: **EchoSolo** (a standalone **X-ESNs** without any base model), **EchoFormer** (**X-ESNs** combined with PatchTST), **EchoMLP** (**X-ESNs** paired with an MLP), **X-ESNs** (**X-ESNs** integrated with TPGN), and **EchoLinear** (**X-ESNs** alongside DLinear). This flexibility enables the framework to adapt across diverse modeling paradigms and datasets, consistently delivering improved performance.

**Results. EFNs** achieves state-of-the-art performance across a range of multivariate TSF benchmarks. For example, on the DMV dataset, **EchoFormer** attains up to a **57.1% relative error reduction** compared to PatchTST, a leading Transformer-based model, as well as other state-of-the-art methods. This demonstrates **EFNs'** ability to capture deep temporal dependencies while maintaining linear computational complexity, with performance gains that increase as the forecasting horizon extends by leveraging the entire observed history.

## 2 ECHO FLOW NETWORKS

We enhance ESNs with scalar-value embedding, group **X-ESNs**, and a MLP readout with cross-attention combination. Each component is detailed below, followed by the complete algorithm.

### 2.1 BACKGROUND: TSF TASK AND BASE MODEL

In a rolling forecasting scenario with fixed context window $k$, the TSF goal is to predict future values $\hat{\mathbf{u}}_{t+1:t+\tau}$ from a short input sequence $\mathbf{u}_{t-k+1:t}$, here a time series dataset is denoted by $\mathbf{u}_{1:T}$ with $\mathbf{u}_t \in \mathbb{R}^{N_u}$. TSF task is expressed as:

$$\hat{\mathbf{u}}_{t+1:t+\tau} = \mathbb{M}(\mathbf{u}_{t-k+1:t}) \tag{2}$$

Here, $\mathbb{M}(\cdot)$ is a TSF **base model**, such as PatchTST, which serves both as a baseline to generate $\hat{\mathbf{u}}_{t+1:t+\tau}$ for comparison and as a component to generate $\hat{\mathbf{u}}'_{t+1:t+\tau}$ in Equation 9 for improvement.

### 2.2 SCALAR-VALUE EMBEDDING

**Embedding Encoder (EE)** To enhance semantic representation in time series forecasting, we adopt SCaNE (Huang et al., 2024) to map each scalar input $u_t \in \mathbb{R}^{N_u}$ into a dense vector, similar to word embeddings. This allows semantically similar values (e.g., 0°F and 100°F in traffic prediction) to be closer in the embedding space. The embedding process is defined as:

$$\mathbf{h}_t = \epsilon(\mathbf{u}_t), \quad \mathbf{h}_t \in \mathcal{R}^{E \times N_u}, \tag{3}$$

where $\epsilon(\cdot)$ is the **scalar-value embedding** function, and $E$ is the embedding dimension.

**Embedding Restoration Decoder (ERD)** After prediction, a restoration decoder maps the high-dimensional embedding outputs $\hat{\mathbf{u}}'_{t+1:t+\tau} \in \mathbb{R}^{EN_u}$ back to the original scalar space $\hat{\mathbf{u}}_{t+1:t+\tau} \in \mathbb{R}^{N_u}$ using a single-layer feedforward network (FFN) with RLEU activation:

$$\hat{\mathbf{u}}_{t+1:t+\tau} = \tilde{\epsilon}(\hat{\mathbf{u}}'_{t+1:t+\tau}) = \text{FFN}(\hat{\mathbf{u}}'_{t+1:t+\tau}) \tag{4}$$

Here, FFN denotes the one-layer feedforward decoder used for dimensionality reduction, and $\hat{\mathbf{u}}'_{t+1:t+\tau}$ is the predictor output before restoring the embedding.

### 2.3 MATRIX-GATED COMPOSITE RANDOM ACTIVATION IN EXTENDED ESNS

To address the limited expressiveness and instability of classical ESNs, as defined in Equation 1, we introduce **Extended Echo State Networks (X-ESNs)**, which incorporate a novel mechanism called **Matrix-Gated Composite Random Activation (MCRA)**, which consists of three key elements: matrix-valued leaky parameters, cascaded composite activations, and randomized heterogeneous activation functions.

**Matrix-Gated Leaky Integration** Classical ESNs use a scalar leaky parameter $\alpha$ to interpolate between the current input and previous states. We generalize this by replacing $\alpha$ with diagonal matrices $\mathbf{W}_1$ and $\mathbf{W}_2$, which gate the contribution of the previous state and current input, respectively. Unlike scalar mixing that only stretches and shrinks in the span, matrix-based gating supports a broader class of linear transformations, including scaling, rotation, shearing, and reflection, allowing state updates to move beyond the original span of input vectors, increasing learning capability.

**Cascaded Composite Activations** Specifically, we apply a pair of nonlinear functions $(\sigma_1, \sigma_2)$ in a nested manner to replace the single `tanh` in ESNs. This cascaded design increases representational capacity by stacking transformations: similar to how simple LEGO blocks can be stacked to form complex structures, this design enables neurons to approximate more complex, hierarchical dynamics. The result is a richer, neuron-specific control over memory decay and update strength.

**Randomized and Heterogeneous Nonlinearities** To further increase diversity and reduce overfitting, we introduce randomness into the choice of activation functions. Instead of a uniform `tanh` across all neurons in ESNs, we randomly assign each **X-ESNs** unit a pair of nonlinearities $(\sigma_1, \sigma_2)$

drawn from a predefined set (e.g., `tanh`, `sigmoid`, `ReLU`, `leaky ReLU`). This ensemble of heterogeneous reservoirs, each with a unique activation signature, allows the model to capture a broader range of temporal dynamics and mitigates the risk of neuron inactivation or gradient vanishing. To bound activations for numerical stability, we apply normalization (i.e., LayerNorm) and clipping.

**MCRA State Update**    Formally, the state update in **X-ESNs** of Equation 2 is refined as:

$$\mathbf{x}_t = \sigma_2 \left( \mathbf{W}_1 \mathbf{x}_{t-1} + \mathbf{W}_2 \cdot \text{Clip} \left( \sigma_1 \left( \text{Norm} \left( \mathbf{W}_{\text{in}} \mathbf{h}_t + \boldsymbol{\theta} + \mathbf{W} \mathbf{x}_{t-1} \right) \right), -1, 1 \right) \right), \tag{5}$$

where $\mathbf{x}_t \in \mathbb{R}^{N_r}$ is the reservoir state, $\mathbf{h}_t \in \mathbb{R}^{N_u}$ is the input embedding, and $\mathbf{W}_{\text{in}}, \mathbf{W}, \boldsymbol{\theta}$ are input, recurrent, and bias weights. $\mathbf{W}_1$ and $\mathbf{W}_2$ are matrix-valued leaky gates. The inner and outer activations $(\sigma_1, \sigma_2)$ are randomly selected from a set of nonlinearities. The classical ESN is recovered when $\mathbf{W}_1, \mathbf{W}_2$ are scalars and $(\sigma_1, \sigma_2) = (\texttt{tanh}, \texttt{linear})$.

## 2.4 MLP READOUT

To enhance expressiveness, we apply an MLP readout to each **X-ESNs**, transforming its internal states into fixed-dimensional, standardized representations. This nonlinear mapping captures rich dynamics while unifying diverse **X-ESNs** outputs to a fixed dimension for grouping:

$$\mathbf{y}_t = \phi_{\text{MLP}}(\mathbf{x}_t) = \sigma(\mathbf{W}_1 \mathbf{x}_t + \boldsymbol{\theta}_1), \quad \mathbf{y}_t \in \mathbb{R}^m \tag{6}$$

Here, $\phi_{\text{MLP}}$ is a learnable nonlinear function implemented as a feedforward neural network, which consists of one fully connected layer with a nonlinear activation function of ReLU $\sigma(\cdot)$. $\mathbf{W}_1 \in \mathbb{R}^{m \times N_r}$ is a learnable weight matrix that maps the **EFNS** with different output dimensions $N_r$ into one unified dimension $m$, and $\boldsymbol{\theta}_1 \in \mathbb{R}^h$ is the bias vector.

## 2.5 GROUP X-ESNS

For stability, to mitigate sensitivity to random initialization of **X-ESNs** state weights, we introduce **Group X-ESNs**, an ensemble approach that reduces prediction variance for more stable outputs. Specifically, we integrate the MLP readouts of multiple independently initialized **Group X-ESNs** to improve stability. We consider $L$ **X-ESNs**, each with distinct decay parameters and output dimension ($\rho$ and $N_r$) as in Gallicchio et al. (2017), leading to a grouped representation $\mathbf{o}_t$ formed by concatenating ($\oplus$) the outputs of all **X-ESNs** readouts:

$$\mathbf{o}_t = \mathbf{y}_t^1 \oplus \mathbf{y}_t^2 \oplus \ldots \oplus \mathbf{y}_t^L \tag{7}$$

### 2.5.1 CROSS ATTENTION COMBINATION

We integrate long-term context from the group **X-ESNs** outputs $\mathbf{o}_t$ (Equation 7) with short-term context from the $k$-window input embeddings $\mathbf{h}_{t-k+1:t}$. This fusion uses a cross-attention operator $\uplus$ that both reads the group **X-ESNs** states relative to the task and merges them with recent short-term inputs. Multiple cross-attention layers are applied sequentially to combine $\mathbf{h}_{t-k+1:t}$ and $\mathbf{o}_t$:

$$\mathbf{h}_{t-k+1:t} \uplus \mathbf{o}_t = \text{LN} \left( \mathbf{o}_t + \text{DO} \left( \text{Softmax} \left( \frac{(\mathbf{o}_t \mathbf{W}^Q)(\mathbf{h}_{t-k+1:t} \mathbf{W}^K)^\top}{\sqrt{d_k}} \right) (\mathbf{h}_{t-k+1:t} \mathbf{W}^V) \right) \right) \tag{8}$$

Here, $\text{LN}(\cdot)$ denotes layer normalization (Ba et al., 2016), which standardizes activations across the hidden dimension, and $\text{DO}(\cdot)$ applies the dropout, a stochastic regularization mask to the attention output (Srivastava et al., 2014). $(\mathbf{W}^Q, \mathbf{W}^K, \mathbf{W}^V)$ are weights of query, key, and value, and $d_k$ denotes the dimensionality of the keys. The fused representation, combining $k$-length context $\mathbf{h}_{t-k+1:t}$ with the **X-ESNs** states $\mathbf{o}_t$, is then passed into a base forecasting model $\mathbb{M}_f$, such as a Transformer-based model. The effective input length remains fixed at $k$, consistent with the baseline input size in Equation 2, and is significantly smaller than the full sequence length ($k \ll T$), ensuring computational efficiency as

$$\hat{\mathbf{u}}'_{t+1:t+\tau} = \tilde{\epsilon} \left( \mathbb{M}_f \left( \mathbf{h}_{t-k+1:t} \uplus \mathbf{o}_t \right) \right). \tag{9}$$

After predictions by $\mathbb{M}_f$, dimensions are restored via $\tilde{\epsilon}$ to yield the final outputs (see Equation 1).

## 2.6 TRAINING ALGORITHM

Algorithm 1 outlines the **EFNS** training procedure. The process begins with parameter initialization. Input sequences are embedded token-wise at each time step (**Step 1**). During each training step, every **X-ESNS** updates its internal states over the full context of length $T$, on non-i.i.d. data without backpropagation (**Step 2**). A fixed MLP readout is then applied to each **X-ESNS**'s state (**Step 3**), and the outputs of all $L$ **X-ESNS** are concatenated to form the Group **X-ESNS** (**Step 4**), which is then fused with the embeddings of the short context with window size $k$ via cross attention (**Step 5**), and are fed into a base model if available (**Step 6**). After that, the embeddings are restored (**Step 7**). The model is trained end-to-end via backpropagation (**Step 8**), by minimizing the standard Huber loss (Meyer, 2021) with details in Appendix A.

---

**Algorithm 1** Training Algorithm

---

**Require:** $\mathbf{u}_{1:T}, \mathbb{M}_f(\cdot), \epsilon(\cdot), \hat{\epsilon}(\cdot)$
**Ensure:** Initialized $\mathbb{M}_f(\cdot)$
    **while** $epoch < epochs$ **do**
        **for** $t \in T$ **do**
            ① **Embedding Encoder**    *[Eq. 3]*
            **for** $l \in L$ **do**
                ② **EFNS**    *[Eq. 5]*
                ③ **MLP Readout**    *[Eq. 6]*
            **end for**
            ④ **Group EFNS**    *[Eq. 7]*
            ⑤ **Joint Attention**    *[Eq. 8]*
            ⑥ **Base Model**    *[Eq. 9]*
            ⑦ **Embedding Restoration**    *[Eq. 4]*
            ⑧ **Huber Loss Backpropagation**
        **end for**
    **end while**

---

## 3 EXPERIMENTS

We test five **EFNS** instantiations without and with different base models: **EchoSolo**: No base model is used; **EchoFormer**: Transformer-based model (PatchTST); **EchoMLP**: an MLP-based model (PatchTSTMixer); **EchoTPGN**: the 2D TSF model (TPGN); and **EchoLinear**: a decomposition-linear model (DLinear) base model, which combined the group **X-ESNS** at the back (see AppendixB).

### 3.1 EXPERIMENTAL SETTINGS

**Evaluation Metrics and Datasets:** We evaluate performance using Mean Squared Error (MSE) and Mean Absolute Error (MAE) (Hastie et al., 2009), where lower values are better. Benchmarks include representative TSF datasets: four ETT variants (ETTh1, ETTh2, ETTm1, ETTm2) (Zhou et al., 2021), Weather and Traffic (Zeng et al., 2022a), Air Quality (AQ) (De Vito et al., 2008), and Daily Website Visitors (DWV) (Nau, 2021). For ETT, Weather, and Traffic, preprocessing follows (Zeng et al., 2023), while AQ and DWV use a 70/10/20 split for training, validation, and testing.

**Baselines:** We compare **EFNS** against a range of strong baselines commonly used in TSF, including Transformer-based models such as PatchTST (Nie et al., 2022), Seg-RNN (Lin et al., 2023), MLP-based models like PatchTSMixer (Ekambaram et al., 2023), linear projection models such as DLinear (Zeng et al., 2023), and large pre-trained models like TimeLLM (Jin et al., 2024). The experimental setup follows the standardized protocol in (Zeng et al., 2023) for fair comparison.

**Model Parameters:** We adopt the hyperparameter settings from Nie et al. (2023); Ekambaram et al. (2023); Toner & Darlow (2024). For PatchTST and PatchTSMixer used as both baselines and base models in **EFNS**, we set FFN dimension to 256, dropout to 0.2, LayerNorm for FFN normalization, a fixed look-back window $k = 336$, patch length 16, stride 8, and 8 layers. For DLinear, we follow its original configuration Toner & Darlow (2024). All models use a learning rate of $1 \times 10^{-3}$ as in Nie et al. (2023). Experiments are conducted on Tesla H100 GPUs using PyTorch Paszke et al. (2019) and HuggingFace Wolf et al. (2020). For large datasets like Weather, embeddings are optionally disabled to reduce memory usage. Full **EFNS** settings are in Appendix K, and variant details in Appendix C.

### 3.2 EVALUATION RESULTS

**Results:** Table 1 shows the TSF prediction results. Statistical significance is computed by running the same algorithm 10 times and averaging the results. Each of our **EFNS** realizations almost always outperforms its corresponding baselines, on three prediction horizons $h \in \{192, 336, 720\}$. **EchoLinear** outperforms DLinear, **EchoMLP** is better than PatchTSMixer, and **EchoFormer** significantly improves over PatchTST and other baselines. Table 2 shows that **EchoFormer** significantly outper-

Table 1: **EFNs** family models compared with baselines including relative improvements (Rel. Imp. %).

| | ETTh1 Dataset | | | | | | DMV Dataset | | | | | |
|---|---|---|---|---|---|---|---|---|---|---|---|---|
| | 192 | | 336 | | 720 | | 192 | | 336 | | 720 | |
| Model | MSE | MAE | MSE | MAE | MSE | MAE | MSE | MAE | MSE | MAE | MSE | MAE |
| DLinear | 0.405 | 0.416 | 0.439 | 0.443 | 0.472 | 0.437 | 0.154 | 0.147 | 0.168 | 0.208 | 0.412 | 0.406 |
| PatchTSMixer | 0.400 | 0.433 | 0.426 | 0.457 | 0.433 | 0.469 | 0.138 | 0.123 | 0.183 | 0.226 | 0.395 | 0.381 |
| PatchTST | 0.379 | 0.256 | 0.435 | 0.462 | 0.447 | 0.472 | 0.142 | 0.106 | 0.172 | 0.212 | 0.427 | 0.404 |
| **EchoFormer** | **0.331** | **0.357** | **0.346** | **0.362** | **0.368** | **0.381** | **0.058** | **0.071** | **0.121** | **0.205** | **0.337** | **0.339** |
| **EchoSolo** | 0.391 | 0.399 | 0.420 | 0.459 | 0.441 | 0.466 | 0.117 | 0.104 | 0.166 | 0.231 | 0.393 | 0.385 |
| **EchoLinear** | 0.407 | 0.409 | 0.426 | 0.430 | 0.436 | 0.428 | 0.087 | 0.117 | 0.142 | 0.222 | 0.388 | 0.412 |
| **EchoMLP** | 0.382 | 0.403 | 0.408 | 0.419 | 0.417 | 0.452 | 0.093 | 0.114 | 0.162 | 0.253 | 0.379 | 0.362 |
| Rel. Imp. % | -19.3 | -15.89 | -19.8 | -17.3 | -15.0 | -13.56 | -57.1 | -31.5 | -20.1 | -1.5 | -14.6 | -12.9 |

Table 2: **EchoFormer** outperforms TSF baselines (horizons: {96, 192, 336, 720}).

| Methods | | EchoFormer (Ours) | | DLinear | | PatchTST | | PatchTsMixer | | Seg-RNN | | TimeLLM | | Rel. Imp. % | |
|---|---|---|---|---|---|---|---|---|---|---|---|---|---|---|---|
| Metric | | MSE | MAE | MSE | MAE | MSE | MAE | MSE | MAE | MSE | MAE | MSE | MAE | MSE | MAE |
| ETTh1 | 96 | **0.331**$_{\pm 0.0225}$ | **0.346**$_{\pm 0.0163}$ | 0.376 | 0.397 | 0.375 | 0.399 | 0.388 | 0.425 | 0.377 | 0.401 | 0.362 | 0.392 | -9.2 | -12.8 |
| | 192 | **0.338**$_{\pm 0.0143}$ | **0.357**$_{\pm 0.0223}$ | 0.405 | 0.416 | 0.413 | 0.421 | 0.400 | 0.433 | 0.422 | 0.441 | 0.416 | 0.425 | -16.5 | -14.2 |
| | 336 | **0.346**$_{\pm 0.0219}$ | **0.362**$_{\pm 0.0152}$ | 0.439 | 0.443 | 0.435 | 0.462 | 0.426 | 0.457 | 0.439 | 0.457 | 0.440 | 0.462 | -18.8 | -21.7 |
| | 720 | **0.368**$_{\pm 0.0227}$ | **0.381**$_{\pm 0.0121}$ | 0.472 | 0.490 | 0.447 | 0.472 | 0.433 | 0.469 | 0.434 | 0.447 | 0.450 | 0.462 | -15.1 | -19.8 |
| | avg | **0.345** | **0.361** | 0.422 | 0.437 | 0.413 | 0.430 | 0.414 | 0.446 | 0.418 | 0.436 | 0.420 | 0.432 | -17.3 | -16.6 |
| ETTh2 | 96 | 0.273$_{\pm 0.0148}$ | **0.291**$_{\pm 0.0137}$ | 0.289 | 0.353 | 0.274 | 0.336 | 0.334 | 0.355 | **0.263** | 0.322 | 0.277 | 0.350 | 3.5 | -9.7 |
| | 192 | **0.293**$_{\pm 0.0144}$ | **0.317**$_{\pm 0.0235}$ | 0.383 | 0.418 | 0.339 | 0.379 | 0.341 | 0.380 | 0.337 | 0.372 | 0.355 | 0.380 | -14.1 | -16.4 |
| | 336 | 0.301$_{\pm 0.0222}$ | **0.321**$_{\pm 0.0236}$ | 0.448 | 0.465 | 0.329 | 0.380 | 0.368 | 0.393 | 0.355 | 0.382 | 0.368 | 0.409 | -8.6 | -15.5 |
| | 720 | **0.327**$_{\pm 0.0121}$ | **0.355**$_{\pm 0.0126}$ | 0.605 | 0.551 | 0.379 | 0.422 | 0.384 | 0.416 | 0.394 | 0.424 | 0.500 | 0.497 | -14.8 | -14.7 |
| | avg | **0.298** | **0.363** | 0.431 | 0.446 | 0.298 | 0.322 | 0.414 | 0.427 | 0.337 | 0.375 | 0.385 | 0.398 | 0 | -3.2 |
| ETTm1 | 96 | **0.283**$_{\pm 0.0214}$ | **0.322**$_{\pm 0.0218}$ | 0.299 | 0.343 | 0.290 | 0.342 | 0.312 | 0.346 | 0.291 | 0.335 | 0.290 | 0.331 | -2.8 | -3.1 |
| | 192 | **0.331**$_{\pm 0.0121}$ | **0.361**$_{\pm 0.0115}$ | 0.335 | 0.365 | 0.332 | 0.369 | 0.348 | 0.374 | 0.366 | 0.381 | 0.347 | 0.369 | -0.5 | -1.1 |
| | 336 | **0.358**$_{\pm 0.0142}$ | **0.372**$_{\pm 0.0339}$ | 0.369 | 0.386 | 0.366 | 0.392 | 0.410 | 0.411 | 0.388 | 0.401 | 0.357 | 0.385 | 0.4 | -3.7 |
| | 720 | **0.401**$_{\pm 0.0241}$ | 0.425$_{\pm 0.0137}$ | 0.425 | 0.421 | 0.416 | 0.420 | 0.405 | 0.418 | 0.412 | 0.418 | 0.409 | 0.436 | -1.0 | 1.6 |
| | avg | **0.338** | **0.368** | 0.357 | 0.378 | 0.351 | 0.380 | 0.400 | 0.406 | 0.362 | 0.389 | 0.429 | 0.425 | -3.8 | -2.7 |
| ETTm2 | 96 | 0.180$_{\pm 0.0213}$ | 0.274$_{\pm 0.0142}$ | 0.187 | 0.269 | 0.165 | 0.255 | 0.167 | 0.267 | **0.158** | **0.241** | 0.170 | 0.277 | 13.9 | 12.4 |
| | 192 | **0.196**$_{\pm 0.0126}$ | **0.254**$_{\pm 0.0132}$ | 0.224 | 0.303 | 0.220 | 0.292 | 0.233 | 0.275 | 0.215 | 0.283 | 0.236 | 0.273 | -11.0 | -6.7 |
| | 336 | **0.279**$_{\pm 0.0147}$ | **0.341**$_{\pm 0.0139}$ | 0.281 | 0.342 | 0.285 | 0.329 | 0.305 | 0.339 | 0.281 | **0.317** | 0.276 | 0.388 | -2.2 | 7.5 |
| | 720 | **0.351**$_{\pm 0.0115}$ | **0.383**$_{\pm 0.0191}$ | 0.397 | 0.421 | 0.362 | 0.385 | 0.408 | 0.403 | 0.357 | 0.391 | 0.362 | 0.388 | -2.7 | -0.6 |
| | avg | **0.247** | **0.315** | 0.267 | 0.333 | 0.255 | 0.315 | 0.278 | 0.321 | 0.253 | **0.306** | 0.347 | 0.293 | -2.4 | 2.9 |
| Weather | 96 | 0.193$_{\pm 0.0219}$ | 0.205$_{\pm 0.0227}$ | 0.176 | 0.237 | **0.149** | **0.198** | 0.172 | 0.220 | 0.158 | 0.203 | 0.153 | 0.281 | 29.5 | 3.5 |
| | 192 | **0.201**$_{\pm 0.0224}$ | **0.241**$_{\pm 0.0226}$ | 0.220 | 0.282 | 0.213 | 0.244 | 0.217 | 0.247 | 0.201 | 0.247 | 0.196 | 0.257 | -5.7 | -1.3 |
| | 336 | **0.235**$_{\pm 0.0322}$ | **0.257**$_{\pm 0.0319}$ | 0.265 | 0.319 | 0.245 | 0.282 | 0.250 | 0.274 | 0.237 | 0.269 | 0.262 | 0.279 | -0.4 | -4.5 |
| | 720 | **0.302**$_{\pm 0.0233}$ | **0.337**$_{\pm 0.0241}$ | 0.333 | 0.362 | 0.325 | 0.357 | 0.319 | 0.339 | 0.311 | 0.348 | 0.304 | 0.356 | -0.7 | -3.2 |
| | avg | **0.230** | **0.257** | 0.248 | 0.300 | 0.235 | 0.264 | 0.259 | 0.287 | 0.226 | 0.264 | 0.271 | 0.334 | -2.2 | -2.7 |
| Traffic | 96 | **0.288**$_{\pm 0.0328}$ | **0.235**$_{\pm 0.0337}$ | 0.410 | 0.282 | 0.360 | 0.249 | 0.367 | 0.357 | 0.543 | 0.255 | 0.388 | 0.264 | -25.0 | -6.6 |
| | 192 | **0.351**$_{\pm 0.0331}$ | **0.256**$_{\pm 0.0328}$ | 0.423 | 0.287 | 0.379 | 0.262 | 0.384 | 0.268 | 0.567 | 0.281 | 0.374 | 0.247 | -7.4 | -2.3 |
| | 336 | **0.362**$_{\pm 0.0328}$ | 0.273$_{\pm 0.0314}$ | 0.436 | 0.296 | 0.392 | **0.269** | 0.393 | 0.268 | 0.602 | 0.307 | 0.385 | 0.271 | -7.7 | 0.1 |
| | 720 | **0.389**$_{\pm 0.0433}$ | **0.281**$_{\pm 0.0423}$ | 0.466 | 0.315 | 0.432 | 0.286 | 0.435 | 0.286 | 0.671 | 0.481 | 0.430 | 0.288 | -10.0 | -1.8 |
| | avg | **0.348** | **0.262** | 0.433 | 0.295 | 0.390 | 0.263 | 0.372 | 0.257 | 0.595 | 0.331 | 0.391 | 0.267 | -11.5 | -0.1 |
| Air Quality | 192 | **0.494**$_{\pm 0.0299}$ | **0.472**$_{\pm 0.0214}$ | 0.569 | 0.601 | 0.541 | 0.538 | 0.511 | 0.535 | 0.526 | 0.566 | 0.519 | 0.526 | -3.5 | -11.3 |
| | 336 | **0.538**$_{\pm 0.0231}$ | **0.519**$_{\pm 0.0178}$ | 0.541 | 0.524 | 0.598 | 0.562 | 0.573 | 0.571 | 0.551 | 0.591 | 0.551 | 0.560 | -0.4 | -7.4 |
| | 720 | **0.678**$_{\pm 0.0315}$ | **0.632**$_{\pm 0.0322}$ | 0.835 | 0.862 | 0.728 | 0.764 | 0.792 | 0.754 | 0.701 | 0.681 | 0.681 | 0.682 | -0.5 | -7.4 |
| | avg | **0.470** | **0.531** | 0.648 | 0.662 | 0.728 | 0.622 | 0.625 | 0.620 | 0.592 | 0.612 | 0.562 | 0.589 | -16.4 | -9.9 |
| DMV | 192 | **0.061**$_{\pm 0.0045}$ | **0.074**$_{\pm 0.0071}$ | 0.154 | 0.147 | 0.142 | 0.106 | 0.138 | 0.123 | 0.135 | 0.117 | 0.149 | 0.113 | -54.9 | -30.2 |
| | 336 | **0.133**$_{\pm 0.0428}$ | **0.198**$_{\pm 0.0628}$ | 0.168 | 0.205 | 0.172 | 0.212 | 0.183 | 0.226 | 0.158 | 0.201 | 0.201 | 0.219 | -15.9 | -5.3 |
| | 720 | **0.352**$_{\pm 0.0219}$ | **0.348**$_{\pm 0.0351}$ | 0.412 | 0.406 | 0.427 | 0.404 | 0.395 | 0.381 | 0.418 | 0.355 | 0.388 | 0.411 | -9.3 | -2.0 |
| | avg | **0.182** | **0.213** | 0.244 | 0.250 | 0.238 | 0.244 | 0.225 | 0.243 | 0.291 | 0.298 | 0.246 | 0.248 | -28.7 | -13.4 |

forms baseline models across most datasets in different horizons. Notably, **EchoFormer** achieves a deduction of MSE on the DMV dataset from 0.138 to 0.061, i.e., -57.1% relatively.

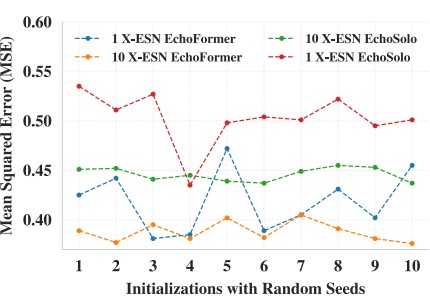

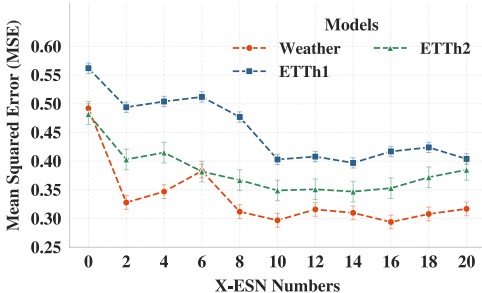

Figure 2: MSE vs. different initializations. Group **EFNs** (e.g., 10 **EFNs**) improve stability and error rates over a single **EFNs** on ETTh1. **EchoFormer** and **EchoSolo** are not sensitive to initialization.

Figure 3: MSE vs. **EFNs** numbers. Optimal **EFNs** Numbers in Group **EFNs** converge around 10 across datasets ETTh1, ETTh2, and Weather.

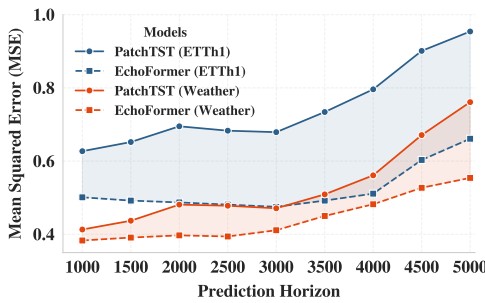 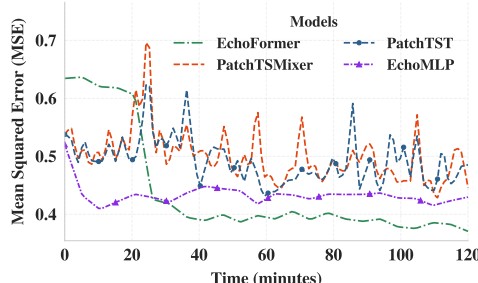

Figure 4: MSE vs. horizon length: **EchoFormer** out-
performs the baseline across all horizons, with widen-
ing margins as the horizon extends, on validation set
(horizon 720 for all ablations by default).

Figure 5: Model training efficiency. **EchoFormer** and
**EchoMLP** converge more quickly and achieve lower
error rates than baselines on the ETTh1 validation set.

Table 3: Training time (min) and memory (GB).

| Dataset | EchoFormer | | EchoSolo | | PatchTST | | DLinear | |
|---|---|---|---|---|---|---|---|---|
| | min | GB | min | GB | min | GB | min | GB |
| ETTh1 | 32 | 12 | 10 | 3 | 40 | 10 | 14 | 4 |
| ETTh2 | 33 | 12 | 9 | 4 | 40 | 10 | 16 | 4 |
| ETTm1 | 57 | 18 | 17 | 6 | 60 | 16 | 25 | 7 |
| ETTm2 | 57 | 19 | 20 | 6 | 60 | 16 | 24 | 8 |
| Weather | 47 | 155 | 15 | 30 | 48 | 130 | 21 | 34 |
| Traffic | 86 | 503 | 25 | 68 | 93 | 453 | 27 | 88 |

**Memory and Time Complexity Comparison:**
Let $N_r$ be the **X-ESNs** state size, $W$ the number
of **X-ESNs**, $k$ the short-term window, $T$ the long-
term input length, $L$ the number of layers, $r$ the
compression ratio, $P$ the patch size, and $d_\epsilon$ the em-
bedding dimension. The time and memory com-
plexities of **X-ESNs** are $O(TN_r^2)$ and $O(N_r^2)$, re-
spectively. For **EchoFormer**, the added cost from

PatchTST yields time complexity $O\left(\frac{T^2 d_\epsilon}{P^2} + N_r T k\right)$ and memory complexity $O\left(\frac{T^2 d_\epsilon}{P^2} + LT^2\right)$.
Despite its richer architecture, **EchoFormer** converges faster and achieves lower MSE than PatchTST,
as shown in Figure 5. Training time, GPU usage, and efficiency metrics are summarized in Table 3.

### 3.3 ABLATION STUDY

**Stability of Group X-ESNs:** Figure 2 compares **EchoFormer** and **EchoSolo** under grouped (10
**EFNS**) and single (1 **EFNS**) configurations. Grouped **X-ESNs** exhibit significantly lower error and
greater stability across runs, indicating reduced sensitivity to initialization.

**Effect of Group Size:** As shown in Figure 3, increasing the number of **X-ESNs** in the group leads to
consistent improvements in MSE, with diminishing returns beyond size 10.

**Forecast Horizon Robustness:** Figure 4 shows our method outperforms baselines across all predic-
tion horizons. The performance gap increases with longer horizons, demonstrating strong long-term
forecasting capabilities.

Table 4: Component effectiveness on ETTh1.

**Component Effectiveness:** Table 4 reports
ablation results on ETTh1, divided into three
sections. The first shows baseline perfor-
mance of PatchTST and TPGN. The second
analyzes the impact of modifying or remov-
ing EchoFormer components. The third starts
from classic ESNs Jaeger et al. (2007) and
incrementally builds up with **EFNS**.

In the second section, removing scalar value
embedding and its restoration causes a sig-
nificant performance drop. Replacing cross-
attention with concatenation or averaging also
degrades results, confirming cross-attention's
superiority in fusing multi-source information.
Increasing levels in the cascaded nonlinear ac-

| | Model | MSE | MAE |
|---|---|---|---|
| | **PatchTST** | 0.447 | 0.472 |
| | **TPGN** | 0.519 | 0.541 |
| | **EchoFormer** | 0.368 | 0.381 |
| | **Without Embedding** | 0.465 | 0.490 |
| | **Cross Attention → Concatenation** | 0.612 | 0.673 |
| | **Cross Attention → Average** | 0.488 | 0.519 |
| | **3 Level Composite** | 0.371 | 0.390 |
| | **4 Level Composite** | 0.368 | 0.384 |
| | **All MCRA Activation: ReLU** | 0.392 | 0.401 |
| | **All MCRA Activation: Sigmoid** | 0.372 | 0.382 |
| | **All MCRA Activation: tanh** | 0.375 | 0.390 |
| 1 | **ESN** | 0.681 | 0.657 |
| 2 | **1 + MLP Readout** | 0.627 | 0.606 |
| 3 | **2+ Group ESNs + Cross Attention** | 0.507 | 0.531 |
| 4 | **3+ tanh → Random** | 0.482 | 0.539 |
| 5 | **4+ MCRA (EchoSolo)** | 0.441 | 0.466 |
| 6 | **5+ TPGN (EchoTPGN)** | 0.462 | 0.481 |
| 7 | **5+ PatchTST (EchoFormer)** | 0.368 | 0.381 |

tivation function yields no further gain. Replacing randomized activations in **Group X-ESNs** with
a single fixed function also reduces performance. Together, these results show each component
contributes uniquely and synergistically to EchoFormer's improvement.

The third section uses system IDs in the first column to track incremental additions starting from classic ESNs (ID 1). The original ESN has high error, while adding an MLP readout (ID 2) reduces MSE. Further improvements come from Group ESNs and cross-attention, showcasing the power of the dual-stream and ensemble. Substituting `tanh` with randomized activations further boosts performance. Integrating all **X-ESNs** components significantly lowers MSE (a 6.6% reduction). Combining **X-ESNs** with a moderate backbone like TPGN improves TPGN but is worse than EchoSolo, while a stronger backbone like PatchTST leads to the best overall performance in EchoFormer.

## 4 RELATED WORK

**Deep Learning for TSF** Recent TSF models span CNNs (e.g., MICN (Wang et al., 2023), Times-Net (Wu et al., 2023), ModernTCN (Luo & Wang, 2024)) for local patterns; RNNs (e.g., SegRNN (Lin et al., 2023), WITRAN (Jia et al., 2023)) for sequential modeling but prone to gradient issues; and linear models (e.g., FITS (Xu et al., 2024), SparseTSF (Lin et al., 2024b), CycleNet (Lin et al., 2024a)) which are efficient but less expressive. Transformers (e.g., PatchTST (Nie, 2023), TiDE (Huang, 2023), FiLM (Wang, 2023), BasisFormer (Zhou, 2023), iTransformer (Liu et al., 2024), Leddam (Yu et al., 2024)) achieve high accuracy but with high cost. PatchTST (Nie, 2023) and SegRNN (Lin et al., 2023) perform well but struggle with fine-grained or irregular inputs (Nie, 2023; Lin et al., 2023). TimeKAN (Huang et al., 2025) prioritizes efficiency, while TPGN (Jia et al., 2024) risks future leakage (Chi, 2024). **EFNS** mitigates these by learning full history without backpropagation through time, combining accuracy with scalability.

**ESN Efficiency and Stability** ESNs avoid vanishing gradients by using fixed recurrent weights (Jaeger, 2001), requiring no backpropagation through time. They offer constant memory and optimization costs ($\mathcal{O}(1)$) via lightweight readouts. Approaches such as spectral radius tuning (Lukoševičius & Jaeger, 2009), regularization (Rodan & Tiňo, 2011), and reservoir optimization (Lu et al., 2017) aim to stabilize ESNs but often need careful tuning. **EFNS** instead aggregates multiple randomly initialized Group X-ESNs and integrates them via cross-attention—more adaptive than prior parallel reservoirs (Sun et al., 2024; Casanova et al., 2023; Rodan & Tino, 2011). Furthermore, ESNs suffer from initialization sensitivity and limited readout expressiveness. Prior work explores spectral tuning (Jaeger, 2001; Rodan & Tiňo, 2011), nonlinear readouts (Gauthier et al., 2021a), attention (Ma et al., 2022), and graph-based methods (Liao, 2023). **EFNS** unifies ESNs with a dual-stream architecture and attention-based readout for improved accuracy and adaptability.

**Expressive Readouts and Modular Architectures** Classic ESNs use linear readouts (Jaeger et al., 2007; Jaeger, 2001), while recent works explore quadratic (Gauthier et al., 2021b), kernel (Hermans & Schrauwen, 2010), or attention-based mappings (Köster & Anandkumar, 2025), often trading efficiency for expressiveness. **EFNS** outperforms in both accuracy and efficiency. Additionally, although ESNs have been embedded into neural architectures (Sun et al., 2024; Xu et al., 2025; Casanova et al., 2023; Shen et al., 2020), they typically serve fixed roles; **EFNS'** cross-attention readout enables flexible model compositions, yielding multiple effective variants.

## 5 CONCLUSION

**ECHO FLOW NETWORKS** is a powerful and flexible framework for capturing long-term temporal dependencies, consistently improving performance both standalone and as an enhancer to existing TSF models. This work is the first to demonstrate that ESNs can enhance Transformer-based TSF models, highlighting a promising direction for efficient, scalable sequence modeling. We hope this advances future research on ESN-based architectures in accuracy and scalability.

## 6 IMPACT STATEMENT

This work improves TSF with social values.

**Reproducibility statement** **EFNS'** source code is in the supplementary materials. Experimental details and dataset info are in Section 5 and Appendix K.

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

# Appendices

## A   TRAINING LOSS

Our loss function (Meyer, 2021) is as follows:

$$\mathcal{L}(\bar{\mathbf{u}}_{t+1:t+\tau}, \hat{\mathbf{u}}_{t+1:t+\tau}) = \frac{1}{\tau} \sum_{i=t+1}^{t+\tau} \ell(\bar{\mathbf{u}}_i, \hat{\mathbf{u}}_i) \tag{10}$$

$$\ell(\bar{\mathbf{u}}_i, \hat{\mathbf{u}}_i) = \begin{cases} \frac{1}{2}(\bar{\mathbf{u}}_i - \hat{\mathbf{u}}_i)^2, & \text{if } |\bar{\mathbf{u}}_i - \hat{\mathbf{u}}_i| \le \delta \\ \delta\left(|\bar{\mathbf{u}}_i - \hat{\mathbf{u}}_i| - \frac{1}{2}\delta\right), & \text{otherwise} \end{cases}$$

Here, $\delta$ sets the differentiability threshold; $\bar{\mathbf{u}}_i$ is the label and $\hat{\mathbf{u}}_i$ the prediction at time $i$.

As an example, we illustrate how **X-ESNs** outputs are paired with embedded inputs when the look-back window is set to $k = 2$. At each step, the **X-ESNs** state is updated by incorporating the current input $h_t$ into the previous state $x_{t-1}$, yielding the new state $x_t$. For instance, $x_1$ is initialized from $h_1$, $x_2$ is obtained by reading $h_2$ into $x_1$, and $x_3$ is obtained by reading $h_3$ into $x_2$. With $k = 2$, future predictions $\hat{\mathbf{u}}_{4:4+\tau}$ are generated via cross-attention between step-3 states ($x_3$) and the inputs $h_{2:3}$.

For the next time step, we read the $h_4$ value into step 3's states and generate step 4's states. By using the cross attention between step 4's states after linear transformation and $h_{3:4}$, we can make our future prediction $\hat{\mathbf{u}}_{5:5+\tau}$.

## B   ECHO FLOW NETWORKS MODEL ARCHITECTURES WITH FRONT AND END COMBINER

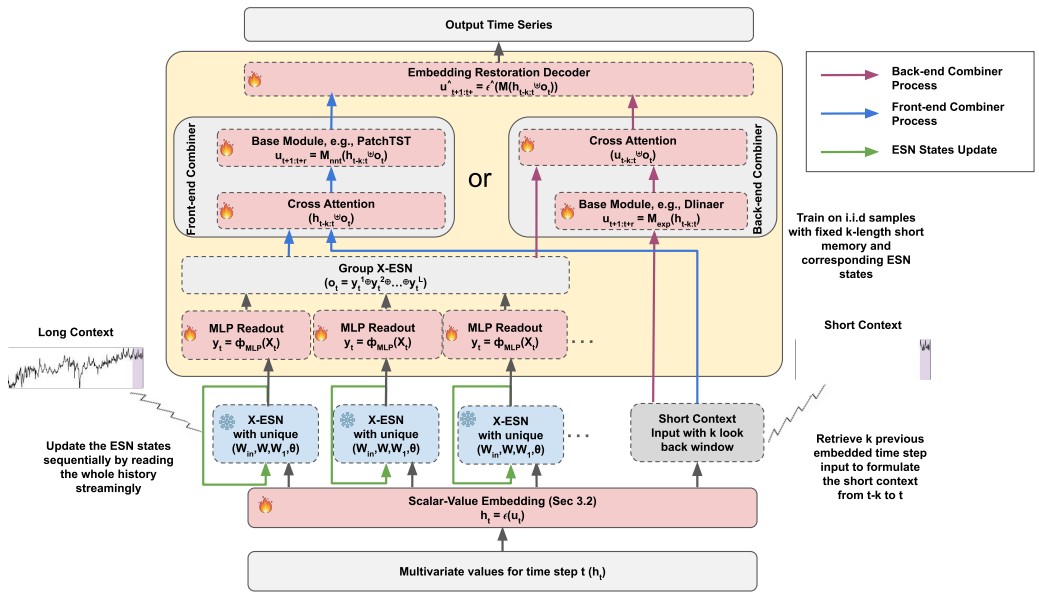

Figure 6: **ECHO FLOW NETWORKS** Models Structure. The components in red are trainable, and those in blue are frozen. Meanwhile, the Green arrows for Reservoirs represent the reservoir state updating process, current reservoir state will be one input for the Reservoir in the next time step. The blue and pink arrow represent the Front-end (Front) and Back-end Combiner.

## C   ECHO FLOW NETWORKS FAMILY DESCRIPTION

- **EchoSolo**: **EchoSolo** use the standalone group **X-ESNs** as a TSF predictor, consisting of the Scaler-Value Embedding, Group **X-ESNs** and Front-end (Front) Combiner. **EchoSolo** uses a Front-end (Front) Combiner, combining the **X-ESNs** output with the actual time series input. Then the outputs are fed into the Embedding Restoration Decoder to restore their shape back to the input shape before the Scaler-Value Embedding to generate the final prediction outputs.

- **EchoFormer**: **EchoFormer** combines Group **X-ESNs** and the Transformer-based PatchTST TSF model. Specifically, **EchoFormer** consists the Scalar-Value Embedding, Group **X-ESNs** and Front-end (Front) Combiner. Following Equation 8, then the results from the Front-end (Front) Combiner are fed into a PatchTST model following the setting pointed in the Model Parameters section. Then, the outputs of the PatchTST model are feed into the Embedding Restoration Decoder to shift the dimension of these outputs back into their original input dimension to generate the final prediction outputs.

- **EchoMLP**: **EchoMLP** combines Group **X-ESNs** and MLP-based PatchTSTMixer. Specifically, **EchoMLP** also include the Scalar-Value Embedding, Group **X-ESNs** and Front-end (Front) Combiner, then the results from the Front-end (Front) Combiner are fed into a PatchTSMixer model following the settings in the Model Parameters section. In the end, the PatcHTSMixer model predictions are shifted by Embedding Restoration Decoder to the original input shapes and generate the final outputs.

- **EchoLinear**: **EchoLinear** combines Group **X-ESNs** with decomposition and linear modeling method of DLinear. Unlike previous models, **EchoLinear** model applied the Back-end (Back) Combiner, the time series inputs first go through the Scaler-Value Embedding layer to expand the dimension of the inputs, then the embedded inputs are fed into the Dlinear model to generate the time series predictions. These predictions from Dlinear model are then corrected and improved by the **X-ESNs** outputs through the Back-end (Back) Combiner explained in Equation 8. The Back-end (Back) Combiner here effectively injects long-range dependencies into Dlinear model's outputs. Finally, the Embedding Restoration Decoder reconstructs the combined high-dimensional results back to the original prediction space and generate the final predictions.

- **EchoTPGN**: **EchoTPGN** integrates an Extented Echo State Network (**EFNs**) with a convolutional network-based TPGN model. Similar to **EchoLinear**, we employ the Back-end (Back) Combiner for TPGN. However, to maintain the effectiveness of the convolutional layers within TPGN, the model inputs are fed directly to TPGN without scaler values embedding. The future predictions generated by TPGN are then combined with the group **X-ESNs** results via cross-attention layers, stabilizing the overall predictions. Crucially, **EchoTPGN** preserves the original input shape and structure required by the TPGN model, maximizing the utility of its internal convolutional layers.

## D   FRONT AND BACK COMBINER

For cross-attention readout, we consider two distinct settings. For neural network-based base models, such as PatchTST, we employ a front combiner to leverage the features learned by the **X-ESNs** groups. In contrast, frequency analytical base models like DLinear, we use a back combiner that directly exploits the inherent structure of the input data to extract features, as follows:

$$\hat{\mathbf{u}}'_{t+1:t+\tau} = \begin{cases} \tilde{\epsilon}\Big(\mathbb{M}_f\big(\mathbf{h}_{t-k+1:t} \uplus \mathbf{o}_t\big)\Big) & \text{Front} \\ \tilde{\epsilon}\Big(\mathbb{M}_b\big(\mathbf{h}_{t-k+1:t}\big) \uplus \mathbf{o}_t\Big) & \text{Back} \end{cases} \tag{11}$$

**Front**   We use cross attention to combine $\uplus$ the embedded input $\mathbf{h}_{t-k+1:t}$ and the group **X-ESNs** output $\mathbf{o}_t$ and then feed into the base model like PatchTST and generate the prediction $\hat{\mathbf{u}}'_{t+1:t+\tau}$ through:

$$\hat{\mathbf{u}}'_{t+1:t+\tau} = \tilde{\epsilon}\Big(\mathbb{M}_f\big(\mathbf{h}_{t-k+1:t} \uplus \mathbf{o}_t\big)\Big) \tag{12}$$

The resulting effective input size to the base model is the same as the baseline input size as in the baseline model, which is significantly smaller than the all-time history length, $k \ll T$.

**Back**    Unlike the front combiner, the back combiner integrates the output from the base model like Dlinear, with the Group **X-ESNs** states $\mathbf{o}_t$ following:

$$\hat{\mathbf{u}}'_{t+1:t+\tau} = \tilde{\epsilon}\Big(\mathbb{M}_b\big(\mathbf{h}_{t-k+1:t}\big) \uplus \mathbf{o}_t\Big) \tag{13}$$

The back combiner is designed especially for decomposition-based models like Dlinear etc....

Table 5: Addtional Component Ablation Study

| Models | MSE | MAE |
|---|---|---|
| **EchoFormer** | 0.368 | 0.381 |
| MSE as loss function | 0.362 | 0.395 |
| with Leaky value $\alpha$ | 0.372 | 0.387 |

## E    LEAKY WEIGHTS AND VALUES COMPARISON

Table 5 also shows that changing the leaky value mechanism to the leaky weights mechanism yields significant improvements in our model, from about $0.372$ to about $0.368$.

## F    LOSS FUNCTION COMPARISON

Table 5 shows that although using MSE as a loss function can reach a slightly lower MSE level, the MAE performance is not ideal compared to using Huber loss as a loss function, which effectively ensures the convergence of both MAE and MSE during training.

## G    EXTENDED ECHO STATE NETWORK SIZE ANALYSIS

Our optimized **X-ESNs** state size is $(100, 105, 110, 115, 120, 125, 130, 135, 140, 145, 150)$ and the number of **X-ESNs** is 10 as in Figure 7 and Figure 3 for all **ECHO FLOW NETWORKS** family models. The optimized spectral radius values are $(0.9, 0.85, 0.80, 0.75, 0.7, 0.65, 0.6, 0.55, 0.5, 0.45)$ for each **X-ESNs** in the group.

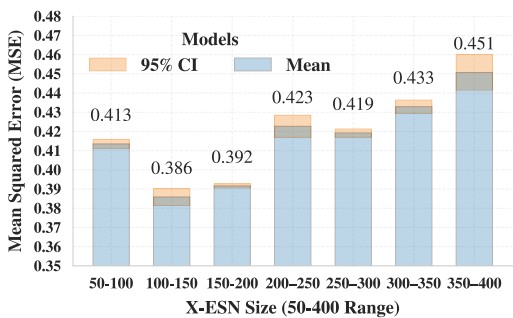

Figure 7: MSE vs. **X-ESNs** size.

Figure 7 shows that **X-ESNs** size significantly affects performance. Testing sizes from 50 to 100 reveals an optimal range of 100–150, roughly $0.8 \sim 1.0 \times E$. Smaller **X-ESNs** lack capacity; larger ones overfit. Proportional sizing to the embedding dimension is key for stable learning.

Table 6: Detailed hyperparameter settings for the group **X-ESNs**

| X-ESNs ID | X-ESNs Size | Spectral Radius | Sparsity | MCRA |
|---|---|---|---|---|
| **X-ESNs**-1 | 100 | 0.90 | 0.60 | Tanh |
| **X-ESNs**-2 | 105 | 0.85 | 0.55 | Sigmoid |
| **X-ESNs**-3 | 110 | 0.80 | 0.50 | Relu |
| **X-ESNs**-4 | 115 | 0.75 | 0.45 | Tanh |
| **X-ESNs**-5 | 120 | 0.70 | 0.40 | Sigmoid |
| **X-ESNs**-6 | 125 | 0.65 | 0.35 | Relu |
| **X-ESNs**-7 | 130 | 0.60 | 0.30 | Tanh |
| **X-ESNs**-8 | 135 | 0.55 | 0.25 | Sigmoid |
| **X-ESNs**-9 | 140 | 0.50 | 0.20 | Relu |
| **X-ESNs**-10 | 145 | 0.45 | 0.15 | Tanh |

## H  COMPLEXITY RANKING OF TIME SERIES FORECASTING METHODS

Table 7: Complexity of time series forecasting methods with theoretical training, practical training (frozen parts), inference, and memory costs. RNN (LSTM/GRU) Hochreiter & Schmidhuber (1997); Cho et al. (2014) Transfer Learning (Frozen Base) Borovykh et al. (2022). Meta-Learning (HyperNet) von Oswald et al. (2019). Foundation Model + Adapter (TimeGPT) Wu (2023b). Frozen Transformer Encoder Li & Wang (2022)
.

| Rank | Method | Trainable Part | Training Time Complexity (full model) | Practical Training Time (frozen parts) | Inference Time |
|---|---|---|---|---|---|
| 1 | ESN Jaeger (2001) | Output layer only | $\mathcal{O}(N^2T + N^3)$ | Same as full | $\mathcal{O}(NT)$ |
| 2 | DLinear Zeng et al. (2022b) | Entire model | $\mathcal{O}(DTB)$ | Same as full | $\mathcal{O}(DT)$ |
| 3 | RNN (LSTM/GRU) | Entire model | $\mathcal{O}(DTB)$ | Same as full | $\mathcal{O}(DT)$ |
| 4 | Transfer Learning (Frozen Base) | Head only | $\mathcal{O}(DTB)$ | $\mathcal{O}(D_{head}TB)$ | $\mathcal{O}(L^2H)$ |
| 5 | PatchTSMixer Song et al. (2023) | Entire mixer | $\mathcal{O}(DTB)$ | Same as full | $\mathcal{O}(DT)$ |
| 6 | PatchTST Nie et al. (2022) | Entire transformer | $\mathcal{O}(L^2HB)$ | Same as full | $\mathcal{O}(L^2H)$ |
| 7 | Frozen Transformer Encoder | Head only | $\mathcal{O}(D_{head}TB)$ | $\approx 0$ (frozen encoder) | $\mathcal{O}(L^2H)$ |
| 8 | Meta-Learning (HyperNet) | Meta-network | $\mathcal{O}(D_{meta}TBM)$ | Same as full | $\mathcal{O}(D_{generated}T)$ |
| 9 | TPGN Wu (2023a) | Entire GNN + temporal layers | High (graph conv + temporal layers) | Same as full | Moderate-High |
| 10 | TimeGPT) | Adapter/head only | $\mathcal{O}(D_{adapter}TB)$ | $\approx 0$ (frozen backbone) | $\mathcal{O}(L^2H)$ |
| | **Memory Complexity** | | $\approx$ weights + activations + inputs; typically $\mathcal{O}(\text{model size} + L \cdot d_{model})$ for deep models | | |

We show complexity in theory in Table 7 with following notations:

- **Theoretical training time complexity:** Cost when training *all* trainable parameters.
- **Practical training time:** Cost when *no* network part is trained or only small heads/adapters are trained (frozen backbone).
- **Inference time complexity:** Cost for forward pass to produce predictions.
- **Memory complexity:** RAM/GPU memory use to store weights, activations, and inputs.

The variable definitions are as follows:

- $T$: length of the time series (sequence length)
- $B$: batch size during training
- $L$: input context length or window size (for transformers and patch models)
- $H$: number of layers in deep models (e.g., transformer layers)
- $N$: number of neurons in the reservoir (for ESN)
- $D$: number of trainable parameters in a model or specific module (head, adapter, etc.)
- $M$: number of tasks or meta-learning episodes
- $d_{model}$: model hidden dimension size (transformer embedding dimension)

Below are details of each method:

- **ESN** trains only the output layer, making both training and inference efficient.
- **DLinear** and **RNN** train the entire network, so practical training is equal to theoretical.

- **Transfer learning** freezes the backbone and trains a small head, reducing practical training time.
- **Frozen Transformer Encoder** and **Foundation Models** typically have near-zero training cost since the backbone is frozen and only adapters or heads are trained.
- **Meta-learning** requires expensive meta-training but fast adaptation at inference.
- **TPGN** fully trains GNN + temporal components, so practical training is as costly as theoretical.

# I  COMPLEXITY ANALYSIS

| Model | Memory | Time |
|---|---|---|
| **Transformer** | $O(L^2 + Ld)$ | $O(T^2 d_\epsilon)$ |
| **PatchTST** | $O(\frac{L^2}{P^2} + \frac{Ld}{P})$ | $O(\frac{T^2 d_\epsilon}{P^2})$ |
| **Informer** | $O(Ld \log L)$ | $O(Td_\epsilon \log T)$ |
| **Autoformer** | $O(L^2 + Ld)$ | $T^2 d_\epsilon$ |
| **Reformer** | $O(Ld \log L)$ | $O(Td_\epsilon \log T)$ |
| **RNN** | $O(Ld^2)$ | $O(Td_\epsilon^2)$ |
| **ESN** | $O(N_r^2)$ | $O(TN_r^2)$ |
| **Echoformer** | $O(\frac{L^2}{P^2} + \frac{Ld}{P} + kN_r L)$ | $O(\frac{T^2 d_\epsilon}{p^2} + N_r k^2)$ |

Table 8: Memory and time complexity comparison.

Table 9: Per-epoch Training Efficiency comparison

| Models | Training time per Epoch | Parameters | Disk Usages |
|---|---|---|---|
| PatchTST | 1.46 min/epoch | 85,169 | 380GB |
| **EchoFormer** | 3.24 min/epoch | 103,720 | 407GB |
| TPGN | 4.72 min/epoch | 3,004,495 | 550GB |

In Table 8, we compare our methods with several other baseline models on the memory (cache) footprint and time complexity of different models, and the notations are as follows. $N_r$ is the **X-ESNs** states output dimension; $L$ is the number of **X-ESNs** in the group **X-ESNs** ; $k$ is the short-term context window length; $T$ is the long-term context total input length; $ly$ is the number of layers; $H$ is the number of attention heads; $c$ is the Compressive Transformer memory size; $r$ is the compression ratio; $p$ is the number of soft-prompt summary vectors; $v$ is the summary vector accumulation steps; $s$ is the kernel size; $P$ is the patch size; $d_\epsilon$ is the embedding model dimension; $d_k$ is the dimension of keys in attention; and $d_v$ is the dimension of values in attention. The baselines include PatchTST Nie et al. (2023) which reduces sequence length via patches, significantly lowering complexity; Informer Zhou et al. (2021) which uses ProbSparse self-attention, suitable for long sequences; AutoformerWu et al. (2022) which introduces auto-correlation mechanism, but the attention remains $O(L^2)$; Reformer Kitaev et al. (2020) which uses LSH to reduce attention computation complexity; and RNN model which computes multi-variants time series step-by-step, holding low complexity but struggles with long-range dependencies. Since we use PatchTST as our base-model, the memory complexity and time complexity of **EchoFormer** is $O((T/p)^2 \cdot d_\epsilon + N_r \cdot k^2)$ and $O((T/p)^2 \cdot d_\epsilon + L \cdot T^2)$ which looks similar with other Transformer-based model with additional memory and time required for tracking, computing and using **X-ESNs** states.

The specific training time, GPU usage and efficiency is shown in Table 3, Figure 15 and Figure 9. Although **EchoFormer**'s structure is more complex than PatchTST, our experiments demonstrate its faster convergence rate. While requiring marginally more time per epoch, **EchoFormer** achieves significantly greater loss reduction compared to baseline models (Figure 15). Consequently, despite slightly higher GPU memory requirements, the total training time remains comparable to or shorter than existing approaches.

# J  VARIABLE DESCRIPTIONS

We have described the used variables and math notations in Table 10.

| Variable | Description |
|----------|-------------|
| $\mathbb{M}(.)$ | Transformer model |
| $\mathcal{R}(.)$ | Deep **X-ESNs** computing |
| $\mathbf{u}_t$ | Input at $t$ time steps |
| $\mathbf{x}_t$ | **X-ESNs** state at $t$ time steps |
| $\mathbf{y}_t$ | Linear output of **X-ESNs** |
| $\mathbf{h}_t$ | Embedding of time steps |
| $\mathbf{o}_t$ | Ensembling of group **X-ESNs** |
| $T$ | Total time input length |
| $k$ | Look back window |
| $N_r$ | **X-ESNs** size |
| $N_u$ | **X-ESNs** input size |
| $m$ | **X-ESNs** output size |
| $L$ | Number of **X-ESNs** in Group |
| $\tau$ | Forecasting horizons |
| $d_\epsilon$ | Model dimension |
| $d_k$ | Dimension of key in attention |
| $d_v$ | Dimension of key in attention |
| $c$ | Compressive Transformer memory size |
| $r$ | Compression ratio |
| $p$ | Number of soft-prompt summary vectors |
| $s$ | Kernel size |
| $P$ | Patch size |
| $\phi_{MLP}$ | MLP readout layer |
| $W_1$ and $W_2$ | learnable diagonal matrices for **X-ESNs** |
| $\mathbf{\Phi}$ | Transformer's learnable parameters |
| $\mathbf{W}_q$ | Queries weights |
| $\mathbf{W}_v$ | Values weights |
| $\mathbf{W}_k$ | Keys weights |
| $\mathbf{W}_{in}$ | Input-to-**X-ESNs** weight matrix |
| $\mathbf{W}$ | **X-ESNs** weight matrix |
| $\boldsymbol{\theta}$ | Bias-to-**X-ESNs** weight |
| $\mathbf{W}_{out}$ | **X-ESNs**-to-readout weight matrix |
| $\boldsymbol{\theta}_{out}$ | Bias-to-readout weight |

Table 10: Descriptions of all variables used in this paper

## K    **X-ESNs** Settings

Here we list the **X-ESNs** settings for all 10 different **X-ESNs** in Table6.

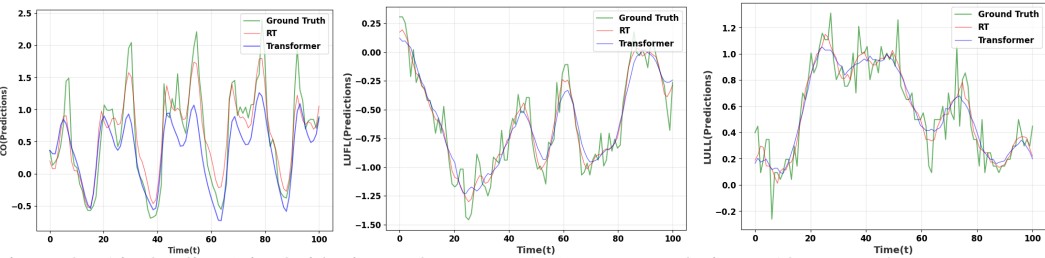

Figure 8: Air Quality (Tin Oxide Readings).

Figure 9: ETTm1 (Low UseFul Load).

Figure 10: ETTm2 (Low UseLess Load).

**Output Examples:** Figure 8, Figure 9, and Figure 10 represent examples on the predicted signal versus the original signal. The cyan line is ground truth; the red line is the prediction using RT; and the blue line is the prediction using Transformer as comparison.

# L    ADDITIONAL COMPONENT ABLATION STUDY

Table 11: Addtional Component Ablation Study

| Models | MSE | MAE |
|---|---|---|
| ESN | 0.681 | 0.657 |
| with MLP readout | 0.627 | 0.606 |
| with Leaky Matrices | 0.621 | 0.598 |
| with Group on sample | 0.619 | 0.592 |
| with Diverse **MCRA** Function | 0.608 | 0.575 |
| with Embedding $\sigma$ | 0.554 | 0.527 |
| with Cross-attention combination ⊎ (**EchoSolo**) | 0.441 | 0.466 |
| with Basemodel (**EchoFormer**) | 0.368 | 0.381 |

# M    CHAOTIC DATASET: LORENZ EXTRACTOR

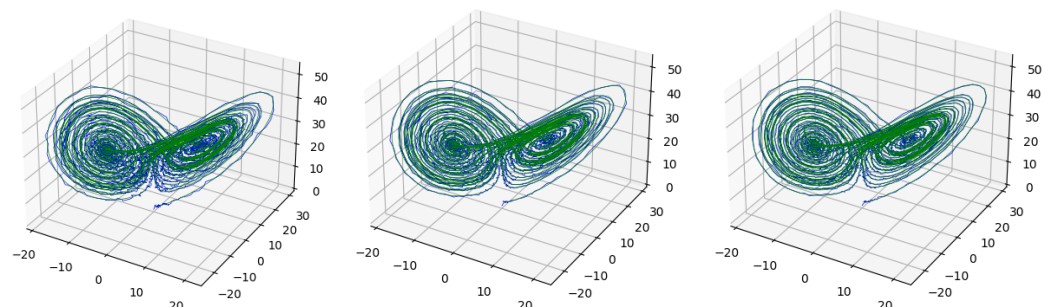

Figure 11: Lorenz with DLinear.    Figure 12: Lorenz with PatchTST.    Figure 13: Lorenz with **ECHO FLOW NETWORKS** .

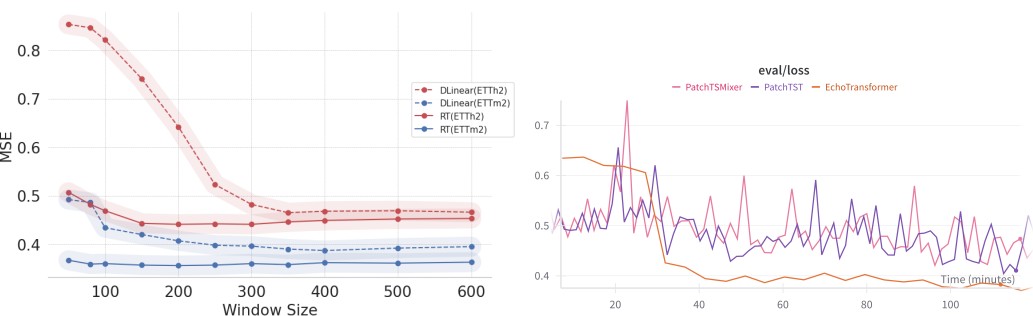

Figure 14: MSE vs. history length.    Figure 15: MSE vs. time.

