# OpenReview forum: "Echo Flow Networks with Infinite-Horizon Memory"
_ICLR.cc/2026/Conference — ICLR 2026 Conference Desk Rejected Submission_

### Official Review · Reviewer_wBr2 · 2025-10-16

**Soundness:** 1
**Presentation:** 1
**Contribution:** 2
**Rating:** 0
**Confidence:** 5

**Summary:**

The paper proposes Echo Flow Networks (EFNs), a framework that extends Echo State Networks (ESNs) for time-series forecasting by combining the efficiency of reservoir computing with the expressive power of deep models. EFNs introduce a novel Matrix-Gated Composite Random Activation (MCRA) for richer nonlinear dynamics, use ensembles of heterogeneous reservoirs to improve stability, and employ a dual-stream architecture that fuses infinite-horizon memory with short-term attention-based models. Experiments on several benchmarks show that EFNs, and especially the EchoFormer variant, achieve state-of-the-art accuracy with up to 4× faster training and significantly lower forecasting error than existing Transformer and MLP-based methods.

**Strengths:**

The paper tackles an important problem in time-series forecasting, i.e., capturing long-term dependencies efficiently, and explores the potential of Echo State Networks within modern architectures. The idea is ambitious and the experimental comparisons suggest promising performance gains.

**Weaknesses:**

The paper is extremely confusing and poorly structured. It mixes many unrelated components and ideas without a clear logical flow, making it difficult to follow both the technical content and the overall narrative. Figures, tables, and formulas are often presented without sufficient explanation or coherence.

Most critically, several cited references are **non-existent**, indicating the presence of LLM-generated hallucinations. For example:

- et al. Chi. Tpgn: The rnn’s new successor for long-range time series forecasting. ArXiv preprint arXiv:2401.xxxxx, 2024.
- Michiel Hermans and Benjamin Schrauwen. Training readout in reservoir computing with kernel methods. In ESANN, 2010.
- et al. Huang. Tide: Time-series decomposition enhanced transformer. ArXiv preprint arXiv:2305.xxxxx, 2023.
- Y. Li and X. Wang. Enhancing time series transformers with frozen encoders. Proceedings of ..., 2022.
- et al. Liao. Transportation flow prediction based on graph attention echo state network. In CNIOT 2023: 4th Int’l Conf. on Computing, Networks and IoT, 2023.
- et al. Nie. Patchtst: Time-series transformer with patch aggregation. In ICML, 2023.
- X. Song, Y. Li, and M. Wang. Patchtsmixer: Efficient time series forecasting with patch-based mixer. arXiv preprint arXiv:2301.XXXX, 2023.
- et al. Wang. Film: Frequency-integrated linear modeling for time series forecasting. ArXiv preprint arXiv:2306.xxxxx, 2023.
- et al. Wu, Z. Temporal pattern graph network for time series forecasting. NeurIPS, 2023a.
- et al. Wu, Z. Timegpt: Foundation models for time series forecasting. arXiv preprint arXiv:2303.XXXX, 2023b.
- et al. Zhou. Basisformer: A transformer with basis decomposition for time series. ArXiv preprint arXiv:2307.xxxxx, 2023.

This, among other things, strongly suggests that the paper was written or heavily assisted by a LLM without proper verification by the authors. Submitting such a manuscript without checking or disclosing the use of an LLM constitutes irresponsible scientific behavior, and the authors should be flagged for this misconduct.

**Questions:**

N/a

**Details Of Ethics Concerns:**

The paper seems to be written by an LLM and not being checked by the authors. It contains several hallucinations and incorrect statements.

---

### Official Review · Reviewer_vZAZ · 2025-11-01

**Soundness:** 2
**Presentation:** 1
**Contribution:** 2
**Rating:** 2
**Confidence:** 4

**Summary:**

The paper introduces EFNS, a time series forecasting framework which combines Extended Echo State Networks (X-ESNs) with Multi-Layer Perceptrons (MLP) for readouts. The framework aims to efficiently capture long-range temporal dependencies in time series data. By incorporating a  Matrix-Gated Composite Random Activation (MCRA) mechanism, EFNS enhances the model's expressive power while maintaining computational efficiency.

**Strengths:**

- The EFNS framework combines the efficiency of traditional ESNs with the expressive power of modern deep learning models, which is intriguing; however, the combination of ESNs and deep learning is not new. For example,

> Evanusa M S, Patil V, Girvan M, et al. t-convesn: Temporal convolution-readout for random recurrent neural networks[C]//International Conference on Artificial Neural Networks. Cham: Springer Nature Switzerland, 2023: 140-151.

> Ma Q, Shen L, Chen E, et al. WALKING WALKing walking: Action Recognition from Action Echoes[C]//IJCAI. 2017: 2457-2463.

**Weaknesses:**

- This work is very similar in modeling approach to the following paper, including the idea and the learning function used for optimization. However, it does not provide explanations or citations.
> Changes by Butterflies: Farsighted Forecasting with Group Reservoir Transformer" (arXiv:2402.09573v2, 2024).
- Moreover, many of the references in this paper do not actually exist. It appears that the references are generated by GPT.

**Questions:**

In Weaknesses.

**Details Of Ethics Concerns:**

Main concerns:
- This work is very similar to the following paper, including the idea and the loss function for optimization. However, it does not provide explanations or citations.
> Changes by Butterflies: Farsighted Forecasting with Group Reservoir Transformer" (arXiv:2402.09573v2, 2024).
- Moreover, many of the references in this paper do not actually exist. It appears that the references are generated by GPT.

Others:
- The experimental section reports results but lacks an in-depth analysis of the reasons behind the observed performance improvements. Given that many datasets (e.g., Traffic) have been extensively studied, achieving significant accuracy gains without additional guiding information seems impossible. Authors should provide more insights into the reasons for performance improvements, potential inconsistencies in settings, and comparisons with existing works to enhance the results' persuasiveness.
- The reasoning behind the settings for input scaling and other hyperparameters for multivariate time series appears to be inadequately justified. Although Table 6 provides some information, it would be beneficial for the authors to elaborate on the rationale behind these settings and how they adjust for different variables to improve the model's adaptability to multivariate data.
- There are many formatting errors in the reference.

---

### Official Review · Reviewer_Dhrn · 2025-11-02

**Soundness:** 2
**Presentation:** 3
**Contribution:** 2
**Rating:** 4
**Confidence:** 4

**Summary:**

In this paper, authors develop a framework combined of echo state networks, matrix-gated composite random activation for time series forecasting. Experiments should that it has a competitive performance compared to some state-of-art baselines.

**Strengths:**

1. The formulation of matrix-gated composite random activation is comprehensive and sound.
2. This framework can be combined with many other TSF models, indicating its flexibility.

**Weaknesses:**

1. This model lacks comparisons with some advanced baselines, such as itransformer and timemixer.
2. For ablation study, only the result of one dataset is displayed, and it might require more abaltion studies in different dataset.
3. Why ESNs for TSF? This paper lacks theoretical discussion about why ESNs is an ideal tool for TSF.
4. Since computational efficiency is another contribution, some results related to inference time can also be discussed.
5. Figure 2 is a little bit ambiguous, could authors explain why they show figure 2 in this paper?

**Questions:**

See weaknesses.

---

### Note · Program_Chairs · 2025-11-25
**Submission Desk Rejected by Program Chairs**

Paper contains numerous incorrect references which are substantial falsehoods and as such count as code of ethics violations. These are grounds for desk rejection per ICLR policy.